# The Role of Magnetic Resonance Imaging in Risk Stratification of Patients with Acute Myocarditis

**DOI:** 10.3390/diagnostics14131426

**Published:** 2024-07-03

**Authors:** Alexandra Popa, Carmen Cionca, Renata Agoston, Flaviu Rusu, Bogdan Mihai Tarcau, Andra Negru, Rares Ilie Orzan, Lucia Agoston-Coldea

**Affiliations:** 1Department of Internal Medicine, Iuliu Hatieganu University of Medicine and Pharmacy, 400347 Cluj-Napoca, Romania; 2Department of Pediatrics, Iuliu Hatieganu University of Medicine and Pharmacy, 400347 Cluj-Napoca, Romania; 3Department of Radiology, Affidea Hiperdia Diagnostic Imaging Centre, 400487 Cluj-Napoca, Romania; 4Faculty of Medicine, Iuliu Hatieganu University of Medicine and Pharmacy, 400347 Cluj-Napoca, Romania; 5Doctoral School of Biomedical Science, University of Oradea, 410087 Oradea, Romania; tarcaubogdanmihai@gmail.com; 6Regional Institute of Gastroenterology and Hepatology “Prof. Dr. Octavian Fodor”, 400162 Cluj-Napoca, Romania; 7Department of Internal Medicine, Emergency County Hospital, 400006 Cluj-Napoca, Romania

**Keywords:** cardiac magnetic resonance imaging, acute myocarditis, left ventricular remodeling, late gadolinium enhancement, cardiac fibrosis, edema, major cardiovascular events

## Abstract

Background: Cardiac magnetic resonance (cMRI) is often used to diagnose acute myocarditis (AM). It is also performed after 6 months to monitor myocardial involvement. However, the clinical and predictive relevance of the 6-month cMRI is uncertain. Objective: We used cMRI to assess the morphology and heart function of patients with AM, the correlation between left ventricular remodeling and biomarkers of heart dysfunction and myocardial fibrosis, and the involvement of myocardial fibrosis initially and 6 months after the acute episode. Materials and methods: We conducted a prospective study of 90 patients with the clinical suspicion of AM, where cMRI was performed within the first week after symptom onset and repeated after 6 months. Results: Non-ischemic late gadolinium enhancement (LGE) was present in 88 (97.7%) patients and mainly involved the septum and inferior wall. cMRI at 6 months was associated with significantly reduced abnormalities of segmental kinetics (*p* < 0.001), myocardial edema (*p* < 0.001), presence of LGE (*p* < 0.05) and LGE mass (*p* < 0.01), native T1 mapping (*p* < 0.001), and presence of pericardial collection (*p* ≤ 0.001). At 6 months, signs of myocardial edema appeared in 34.4% of patients, and a complete cure (absence of edema and LGE) was found in 8.8% of patients. LGE disappeared in 15.2% of patients, and the mean number of myocardial segments involved decreased from 46% to 30%, remaining unchanged in 13% of patients. Patients with LGE without edema had a more severe prognostic condition than those with persistent edema. Patients with increased LGE extension on the control cMRI had a worse prognosis than those with modified or low LGE. The most significant independent predictive parameters for major cardiovascular events (MACEs) were LGE mass (adjusted OR = 1.27 [1.11–1.99], *p* < 0.001), myocardial edema (OR = 1.70 [1.14–209.3], *p* < 0.001), and prolonged native T1 (OR = 0.97 [0.88–3.06], *p* < 0.001). The mid-wall model of LGE and the presence of edema-free LGE were MACE-independent predictors. Conclusions: LGE, myocardial edema, and prolonged native T1 were predictors of MACEs. LGE does not necessarily mean constituted fibrosis in the presence of edema and may disappear over time. LGE without edema could represent fibrosis, whereas the persistence of edema represents active inflammation and could be associated with the residual chance of complete recovery. cMRI should be performed in all patients with AM at 6 months to evaluate progress and prognosis.

## 1. Introduction

Acute myocarditis (AM) is an inflammatory disease of the myocardium. It is clinically underdiagnosed [1,2] and can lead to progressive cardiac remodeling and major cardiovascular events (MACEs). AM is a burden on the health system due to repeated hospitalizations and increased mortality despite therapeutic advances [3]. Myocardial vulnerability in patients with AM competes with the development of myocardial fibrosis, a key mechanism in the progression of cardiac remodeling and the determination of the prognosis of these patients [4]. 

The clinical diagnosis of AM remains challenging due to its non-specific and heterogeneous symptoms, ranging from chest pain, dyspnea, fatigue, or palpitations to brady- and tachyarrhythmias, cardiogenic shock, and sudden cardiac death (SCD) in children and adolescents [5]. Echocardiographic methods, although non-invasive, may not be useful since they lack high sensitivity and specificity in the diagnosis of AM [6]. Endomyocardial biopsy (EMB) was considered the reference standard [7,8], but its sensitivity was not satisfactory [9], and post-procedural complications prevented its widespread application [10].

Cardiac magnetic resonance imaging (cMRI) has proven to be a valuable tool for confirming the diagnosis of AM in the early days of clinical onset through non-invasive evaluation of myocardial inflammation by the presence of edema, hyperemia, and late gadolinium enhancement (LGE) according to the Lake Louise criteria [11]. cMRI allows tissue characterization of the myocardium, as well as the precise definition of volumes and ventricular function. When stratifying cardiovascular risk in patients with AM, only considering cMRI, biomarkers, and EMB for the detection of myocardial fibrosis, cMRI via LGE detects replacement myocardial fibrosis only in 30% of patients [12], providing incremental prognostic value. New T1 and T2 mapping techniques highlight interstitial myocardial fibrosis in up to 88% of patients [13].

CMRI, by characterizing myocardial tissue through LGE and abnormal T2 mapping, provides significant diagnostic and prognostic value for patients with acute coronary syndrome and non-obstructive coronary arteries during angiography [14,15,16,17]. Repeating cMRI at 3 and 6 months has proven useful for evaluating myocardial tissue function and characteristics. The evolution of these patients has been associated with either the normalization of heart function or the progression to dilated cardiomyopathy (DCM), heart failure (HF), or malignant ventricular arrhythmia (VA) and SCD [3]. This suggests that LGE is more a marker of inflammation than of myocardial fibrosis. Some studies have shown that circulating biomarkers of myocardial fibrosis, such as galectin-3 (Gal-3), C-terminal propeptide of type I procollagen (PICP), and N-terminal propeptide of type III procollagen (PIIINP), may provide additional information about the pre-fibrotic state of the myocardium because they reflect collagen turnover [18,19]. Thus, they could help stratify the risk of SCD in these patients.

We aimed to evaluate through cMRI the morphology and heart function in patients with AM, the correlation between the process of left ventricular remodeling and biomarkers of heart dysfunction and myocardial fibrosis, and the involvement of myocardial fibrosis, initially and 6 months after the acute episode.

## 2. Materials and Methods

### 2.1. Patients

We conducted a prospective study of patients with clinical suspicion of AM at the Emergency County Hospital and Hiperdia Medical Imaging Center between October 2018 and November 2021. The inclusion criteria were clinical suspicion of AM, manifested by pseudoanginous or pericardial chest pain, and the presence of at least one of the following changes: (1) a new change on the electrocardiogram, (2) increased troponin levels, or (3) segmental kinetic abnormalities, with left ventricle ejection fraction (LVEF) preserved on ultrasound. Asymptomatic patients were also included if they had suspected AM with at least two of these changes [3]. The exclusion criteria were other types of cardiomyopathies or heart disease, contraindications to cMRI (incompatible metal devices, severe kidney disease with an estimated glomerular filtration rate [eGFR] of <30 mL/min, and claustrophobia), and refusal to participate in the study. 

The certainty of the diagnosis of AM was established based on the Lake Louise cMRI criteria (myocardial edema, hyperemia, and LGE) [3,11]. Once patients with AM were diagnosed, selected, and accepted to participate, they were evaluated according to a standard protocol. Demographic data, comprising age, sex, height, and weight, were documented in addition to the medical history. Cardiovascular symptoms (chest pain, dyspnea, syncope, palpitations) and the current medication regimen were documented. Various biomarkers were assessed, encompassing troponin, leukocyte count, high-sensibility C-reactive protein (hs-CRP), high-sensibility troponin T (hs-troponin T), erythrocyte sedimentation rate (ESR), N-terminal pro-B-type natriuretic peptide (NT-proBNP), Gal-3, and serum creatinine levels. Furthermore, a thorough evaluation of the electrocardiogram in 12 derivatives was conducted to obtain a comprehensive understanding of the cardiac status and any potential abnormalities. Transthoracic echocardiography and cMRI were performed. Ischemic cardiovascular disease was excluded in all patients through either coronary angiography (72 patients) or coronary angioCT (18 patients). Renal function was assessed based on eGFR. Renal dysfunction was defined as an eGFR of <60 mL/min/1.73 m^2^. The data available in the study were supplemented with information from the electronic health sheet and telephone interviews.

The study received approval from the Ethics Commission of the Iuliu Hatieganu University of Medicine and Pharmacy in Cluj-Napoca, and it was conducted following the principles of the Declaration of Helsinki. All patients were informed of the study protocol and signed the informed consent form at the time of the first cMRI examination.

### 2.2. cMRI

The cMRI exploration was performed initially to confirm the diagnosis of AM and subsequently repeated at 6 months after diagnosis using a 1.5 Tesla MRI device (Magnetom Altea, Siemens Medical Systems Solutions, Erlanger, Germany). 

All imaging acquisitions were performed during complete apnea. A standard myocarditis scan protocol was employed for all patients, incorporating steady-state free precession (SSFP) imaging, native T1 and T2 mapping, and LGE [20] (Figure 1).

To assess the function of the left ventricle (LV) and the right ventricle (RV), a rapid imaging protocol was applied using the cine-SSFP sequences in both the longitudinal and short cardiac axes, from the base to the ventricular apex. Based on these acquisitions, the two-, three-, and four-chamber sections of the LV were completed. The acquisition parameters of the cine-SSFP sequences were standardized for optimal blood–cavitary wall discrimination: repetition time (RT) = 3.6 ms, eco-time (ET) = 1.8 ms, angulation angle = 60°, section thickness = 6 mm, field of view = 360 mm, image matrix = 192 × 192 pixels, voxel size = 1.9 × 1.9 × 6 mm, and time resolution = 25–40 ms, at 25 phases of the cardiac cycle.

All images were analyzed offline by two observers with 10 years of experience in the field (LAC and CC). End-diastolic and end-systolic LV volumes (EDV, ESV) and LVEF were measured in the short axis on the cine-SSFP sequences.

Epicardial and endocardial delineations were marked semi-automatically in telesystole and telediastole using the Syngo software (via VB20A_HF04, Argus, Siemens Medical Solutions, Oakville, ON, Canada).

On the T2-weighted images, myocardial edema was considered to be present when the ratio between the intensity of myocardial signal intensity (SI) and that of the skeletal muscle was >2, according to Lok et al. [21]. Myocardial relaxation maps and myocardial relaxation times T1 and T2, as well as extracellular volume fraction values (ECVs), were calculated using a segmentary approach [22,23].

The ECV was calculated using the formula, ECV = (1 − hematocrit) × [∆R1myocardium]/[ΔR1blood-pool], where ∆R1 is the difference between pre- and post-contrast relaxation rates (1/T1), as described by Scully et al. [24]. Abnormally elevated T1 and T2 limits were defined as values >2 standard deviations (SDs) in a healthy population using the same scanners. 

Myocardial hyperemia was assessed using cine-SSFP post-contrast images [25]. Late post-contrast phase-sensitive inversion recovery (PSIR) sequences were aimed at detecting LGE, acquired 10 min after the intravenous administration of 0.2 mmol/kg gadoxetic acid body, using short-axis and longitudinal LV axis sections and reverse-recovery sequences in echo gradient. The measurement and sequence determination parameters for LGE were an RT of 4.8 ms, an ET of 1.3 ms, and an inversion time of 200–300 ms, adjusted to optimally visualize the differentiation of fibrosis from healthy myocardium and black-coded blood pool. Blood pressure from the brachial artery was constantly monitored throughout the acquisition of cine-SSFP images. LGE was qualitatively evaluated for the presence of a non-ischemic distribution pattern (subepicardial or midmyocardial) and as the number of myocardial LGE segments [26]. The LGE scale was measured using a validated method. Endocardial and epicardial contours were drawn manually to identify the LV myocardium in each short-axis image. A region of interest was placed at the level of the myocardium without LGE. LGE was defined as the myocardium with an SI greater than the average SI of the region of concern (>5 SD), without the reference value of the normal myocardium. This SI threshold for LGE has demonstrated the best accuracy and reproducibility in LGE appreciation [27]. The LGE distribution pattern (nodal, subepicardial, midmyocardial, transmural) and the proportion of LV wall damage (0, without LGE; 1, myocardial LGE 0–25% of LV wall thickness; 2, myocardial LGE 26–50% of LV wall thickness; 3, myocardial LGE 51–75% of LV wall thickness; and 4, myocardial LGE 76–100% of LV wall thickness) were assigned to each segment. The myocardial LGE score was calculated for each patient as the sum of segment scores. 

The presence and distribution of edema, hyperemia, and LGE at the LV level were evaluated in the short-axis sections using the 17 segments, as recommended by American College of Cardiology (ACC)/American Heart Association (AHA) [11]. LGE was expressed as a percentage (%) of the LV mass. LGE mass assessment was quantified using the cvi42 software (cvi42 version 5, Circle Cardiovascular Imaging Inc., Calgary, AB, Canada).

### 2.3. Clinical Monitoring

The clinical follow-up was conducted by the patients completing a questionnaire during the initial cMRI session. Monitoring was subsequently performed through clinical visits or via telephone interviews at 1 month, 3 months, and 6 months. The focus was on collecting clinical data related to the occurrence of the first MACE in each patient. A MACE was defined as cardiac death, sustained VT (defined as beats of ventricular origin lasting at least 30 s with a frequency of >100 beats/minute), or HF requiring hospitalization by international guidelines. Extracardiac hospitalizations were not considered during the evaluation.

### 2.4. Statistical Analysis

The normality of the variables was assessed using the Kolmogorov–Smirnov test. The data were presented as mean ± SD, median (25th and 75th percentiles), or percentage. The distribution of nonparametric variables was examined after logarithmic transformation for parametric analysis. Group comparisons used the Chi-square (χ²) or Fischer tests for parametric data and the Kruskal–Wallis H and Wilcoxon tests for nonparametric data, as appropriate. The Pearson correlation coefficient was employed to explore the relationships between variables. 

Logistic regression models were employed to investigate each significant variable identified in the univariate analysis to predict MACE occurrence and whether the adjusted associations between continuous markers of fibrosis and composite results deviated from a linear trend. For adjustment, known predictive covariables for AM, namely, sex and age, were used. A *p*-value of <0.05 was considered statistically significant. 

Inter- and intra-observer coefficients of Cohen’s kappa were calculated. The retrospective calculation of the statistical test’s power and the prospective sample size estimation were conducted using type I and type II variation, dependent on the sample size. The statistical analysis was performed using the MedCalc statistical software (version 19.1.7, MedCalc Software, Ostend, Belgium).

## 3. Results

### 3.1. Basic Features

From the initial study population of 153 patients, patients with suboptimal cMRI images, those who refused to undergo the second cMRI, and those diagnosed with other acute myocardial diseases were excluded (Figure 2). The final study population comprised 90 patients with AM (69 men, average age = 35 ± 14.1 years). 

The main characteristics of patients are presented in Table 1. The most common symptom at presentation was chest pain (91.1%), followed by dyspnea (60%). Prodromal symptoms were observed in 58 (64.4%) patients. Elevated troponin values were found in 89 (98.8%) patients, and hs-CRP was elevated in 79 (87.7%). NT-proBNP levels were elevated in 58 (64.4%) patients, and Gal-3 was elevated in 53 (58.8%) patients. The electrocardiogram revealed at least one abnormality in the vast majority of patients (92.2%).

### 3.2. cMRI Characteristics

The detailed findings of the initial and control cMRIs are summarized in Table 2. The initial cMRI was performed, on average, 5 days (2–7 days) after the onset of symptoms, and the follow-up was conducted at 170 days (136–220 days). In the initial cMRI, the systolic function of the LV was preserved (>50%) in 87 out of 90 patients (96.6%), and the indexed EDV was normal in 88 patients (97.7%). Segmental kinetic abnormalities were present in more than half of the patients (61.1%), and the LV mass was within normal limits for all patients. The systolic function of the RV was preserved in all patients, and the indexed EDV was within normal limits. Pericardial collection was observed in 47.7% of patients.

On the initial cMRI, focal edema on short TI inversion recovery (STIR) images, mainly involving the lateral or inferior wall and septum, was observed in 84 patients (93.3%). Non-ischemic LGE was detected in 88 patients (97.7%) and mainly involved the septum and the inferior wall. The LGE pattern was primarily subepicardial (60%). LV wall damage was predominantly type 1 (0–25% of LV mass) in most cases (60%), followed by type 2 (26–50% of LV mass, 18.8%), type 3 (51–75% of LV mass, 17.7%), and type 4 (76–100% of LV mass, 3.33%). The average LGE mass was 19.7 ± 10.5 g, and the LGE/LV mass ratio was 26.1 ± 9.8%.

Hyperemia was present in 44 (48.8%) patients. Native T1 mapping was increased by more than 1100 ms in 81 patients (90%), and ECV increased by more than 30% in 74 patients (82.2%). Patients with a high LGE mass had higher indexed EDV and lower LVEF.

In the 6-month follow-up cMRI, significant reductions existed in abnormalities of segmental kinetics (*p* < 0.001), myocardial edema (*p* < 0.001), the presence of LGE (*p* < 0.05) and LGE mass (*p* < 0.01), native T1 mapping (*p* < 0.001), and the presence of pericardial collection (*p* ≤ 0.001). Signs of myocardial edema were absent in 34.4% of patients, and a complete cure (absence of edema and LGE) was observed in 8.8%. LGE disappeared in 15.2% of patients, and the number of myocardial segments involved decreased from 46% to 30%, remaining unchanged in 13% of patients. 

Patients with LGE without edema had a more severe prognostic condition than those with persistent edema. Additionally, patients with increased extension of LGE on the control cMRI had a worse prognosis than those with modified or low LGE.

### 3.3. The Ability of Biomarkers to Establish the Diagnosis of Myocardial Fibrosis in AM

The ROC analysis for identifying myocardial fibrosis (Figure 3) demonstrated an area under the curve of 0.807 (95% CI = 0.710–0.882; *p* < 0.0001) for Gal-3. The determined cutoff value was >12.2 ng/dL, yielding a sensitivity of 89.47%, a specificity of 69.23%, a positive predictive value of 68%, and a negative predictive value of 90%.

### 3.4. Determination of Independent Predictors of MACE

During the follow-up period, 15 (16.6%) patients experienced MACEs (OR = 2.9, 95% CI 1.7–7.2), which included documented VT on Holter electrocardiogram (*n* = 3) and clinically manifested HF (*n* = 12). No cardiac deaths were recorded. Higher levels of LGE were associated with HF-type MACEs and VA.

Univariate analysis revealed that clinical HF, chest pain, LGE mass, mass ratio of LGE to LV, prolonged native T1, myocardial edema, LVEF, NT-proBNP, and Gal-3 were all associated with MACEs. A multivariate logistic regression model was constructed, incorporating the significant parameters identified in the univariate analysis. 

The most significant independent predictive parameters for MACEs were LGE mass (adjusted OR = 1.27, 95% CI 1.11–1.99, *p* < 0.001), myocardial edema (OR = 1.70, 95% CI 1.14–209.3, *p* < 0.001), and prolonged native T1 (OR = 0.97, 95% CI = 0.88–3.06, *p* < 0.001; Table 3). The mid-wall LGE in the septum and the presence of edema-free LGE were identified as MACE-independent predictors.

## 4. Discussion

This study integrated cMRI imaging and myocardial biomarkers of dysfunction, fibrosis, and inflammation in patients with AM. It allowed for the independent evaluation of each diagnostic method and assessed their combined value for risk stratification.

At 6 months, complete resolution (absence of edema and LGE) was achieved in a minority of the cases. Patients with LGE but without edema exhibited a more severe prognosis compared to those with persistent edema. Furthermore, patients showing increased extension of LGE on follow-up cardiac MRI had a worse prognosis than those with stable or reduced LGE. The mid-wall septal pattern of LGE and the presence of edema-free LGE were identified as independent predictors of major adverse cardiovascular events (MACEs).

Due to its multiparametric tissue characterization, cMRI is a valuable non-invasive option in clinical practice. It confirms myocardial inflammation and offers prognostic value in everyday practice. cMRI can identify myocardial inflammation irrespective of the underlying etiology, relying mostly on extracellular volume changes to visualize myocardial inflammation [17,28]. At 6 months following symptom onset, our results showed that cardiac edema had resolved in most patients, with only 13% still exhibiting it. During follow-up, both LGE mass and LGE/LV mass ratio significantly decreased. This was likely due to edema re-absorption and reduced inflammatory cell infiltration. Our findings suggest that LGE in AM may not be necessarily irreversible since 13% of patients showed complete LGE disappearance at the 6-month follow-up. In a study by Aquaro et al. [28], similar findings were observed, with 11% of patients achieving full recovery from edema and LGE at the 6-month cMRI and 10% showing total resolution.

In AM, inflammatory cell infiltration and the disease process itself can increase interstitial space, hindering the washout of gadolinium-based contrast medium. The appearance of LGE on cMRI is associated with disorders that augment interstitial space, such as replacement fibrosis, edema, and an excess of proteins. This leads to a delayed washout of gadolinium in the myocardium. During AM, the increase in interstitial space may be attributed to factors such as replacement fibrosis, edema, and the infiltration of inflammatory cells. Interstitial fluids, including gadolinium, might be absorbed into phagosomes when macrophages phagocytize necrotic myocytes. However, these fluids cannot penetrate viable cells with intact membranes. Additionally, inflammatory cells might obstruct lymphatic channels, further reducing gadolinium washout and contributing to interstitial edema [28]. Therefore, in the 6-month follow-up cMRI, LGE is often diminished or even nonexistent, indicating a favorable prognosis. The presence of LGE on cMRI appears to be associated with a worsened prognosis. This underscores the utility of cMRI in not only the diagnosis of AM but also its management.

Another interesting finding from our study is that patients with LGE without edema had a more severe prognostic condition than those with persistent edema. This lack of edema was associated with poor outcomes, aligning with the results of a retrospective analysis involving 388 clinically suspected myocarditis patients [29]. This observation supports the notion that the recovery of LVEF and increased overall myocardial inflammation are essential in patients with myocarditis. In a study involving 187 suspected myocarditis patients, Aquaro et al. [28] reported that individuals with baseline edema fully recovered in the follow-up cMRI after 6 months, consistent with our findings. Acute myocardial edema has not been confirmed in the literature to worsen prognosis. Despite its expansion, acute myocardial edema in clinically suspected or biopsy-proven myocarditis has been associated with improved prognosis. These data collectively suggest that acute myocardial edema, rather than LGE, serves as a sign of reversible damage and a favorable prognosis. Patients with LGE and edema may have active inflammation that could heal, whereas those without edema may exhibit definite fibrosis [30,31].

Our study revealed that the mid-wall septum model and edema-free LGE independently predicted significant adverse cardiac events. Recent findings by Imazio et al. [32] demonstrated that myocardial septal LGE could serve as a predictive factor for patients with AM undergoing cMRI. In their study, septal LGE was observed in 21 (30%) out of 71 patients. Over 60.8 months, the mean LVEF increased, and only four patients (6%) with lower baseline LVEF experienced combined adverse events. Notably, these individuals exhibited septal LGE more frequently. LGE on cMRI has shown its potential to identify high-risk patients with biopsy-proven viral myocarditis, according to Greulich et al. [33]. Mid-wall and septal LGE patterns were strongly associated with mortality, whereas those without LGE demonstrated improved outcomes. Consequently, patients with biopsy-proven myocarditis displaying mid-wall and septal LGE patterns should be carefully monitored in clinical practice, a finding consistent with the study by Gräni et al. [34] and this study.

The study limitations should be acknowledged. Firstly, this was a monocentric study, potentially limiting the general applicability of the findings. Additionally, the recruitment of patients spanned a lengthy period, which may have influenced the quality of the samples for biomarker measurements. Furthermore, the study focused exclusively on patients with AM maintaining an unaltered LVEF and presenting with an infarct-like appearance. Importantly, different findings may arise in individuals with AM and left ventricular dysfunction, HF, or arrhythmia. The decision to enroll patients with unaltered LVEF and an infarct-like presentation was made to capitalize on the precision of cMRI in identifying AM in this specific subgroup. Future research endeavors should extend their investigations to assess the prognostic value of cMRI in patients with HF and arrhythmia.

## 5. Conclusions

LGE, myocardial edema, and prolonged native T1 were predictors of MACEs. The presence of LGE does not necessarily imply established fibrosis in the presence of edema and may resolve over time. LGE without myocardial edema may indicate fibrosis, whereas the persistence of edema suggests active inflammation and could be associated with the residual chance of complete recovery. cMRI should be performed in all patients with AM at 6 months to assess their progress and prognosis.

## Figures and Tables

**Figure 1 diagnostics-14-01426-f001:**
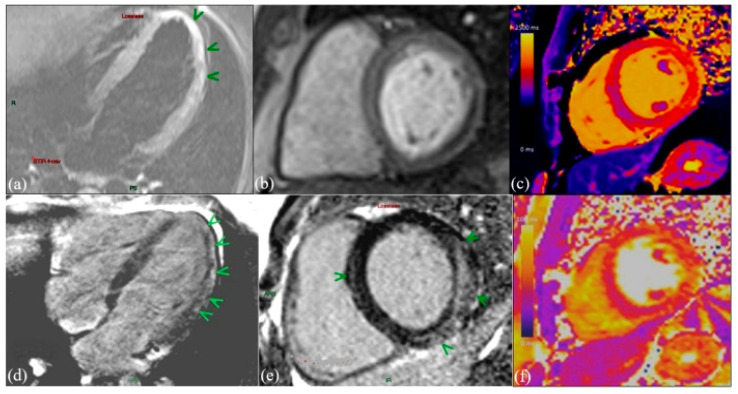
cMRI in a patient with acute myocarditis: (**a**) Long axis of 4 chambers with T2-weighted STIR hyper signal mid-wall and subepicardial inferolateral regions of LV = myocardial edema (green arrows). (**b**) Short axis-T2-weighted STIR-hyperintense subepicardial rim = myocardial hyperemia. (**c**) Short axis-native T1-mapping prolonged T1 relaxation times in the inferolateral regions of the LV (1176 ± 40 msec) = interstitial fibrosis. (**d**) Long axis 4 chambers and (**e**) short axis-late gadolinium enhancement-patchy mid-wall and subepicardial LGE on inferolateral wall of the LV = permanent fibrosis (green arrows). (**f**) Sort axis-native T2 mapping-prolonged T2 relaxation times in the inferolateral regions of the LV (60 ± 14 msec) = myocardial edema.

**Figure 2 diagnostics-14-01426-f002:**
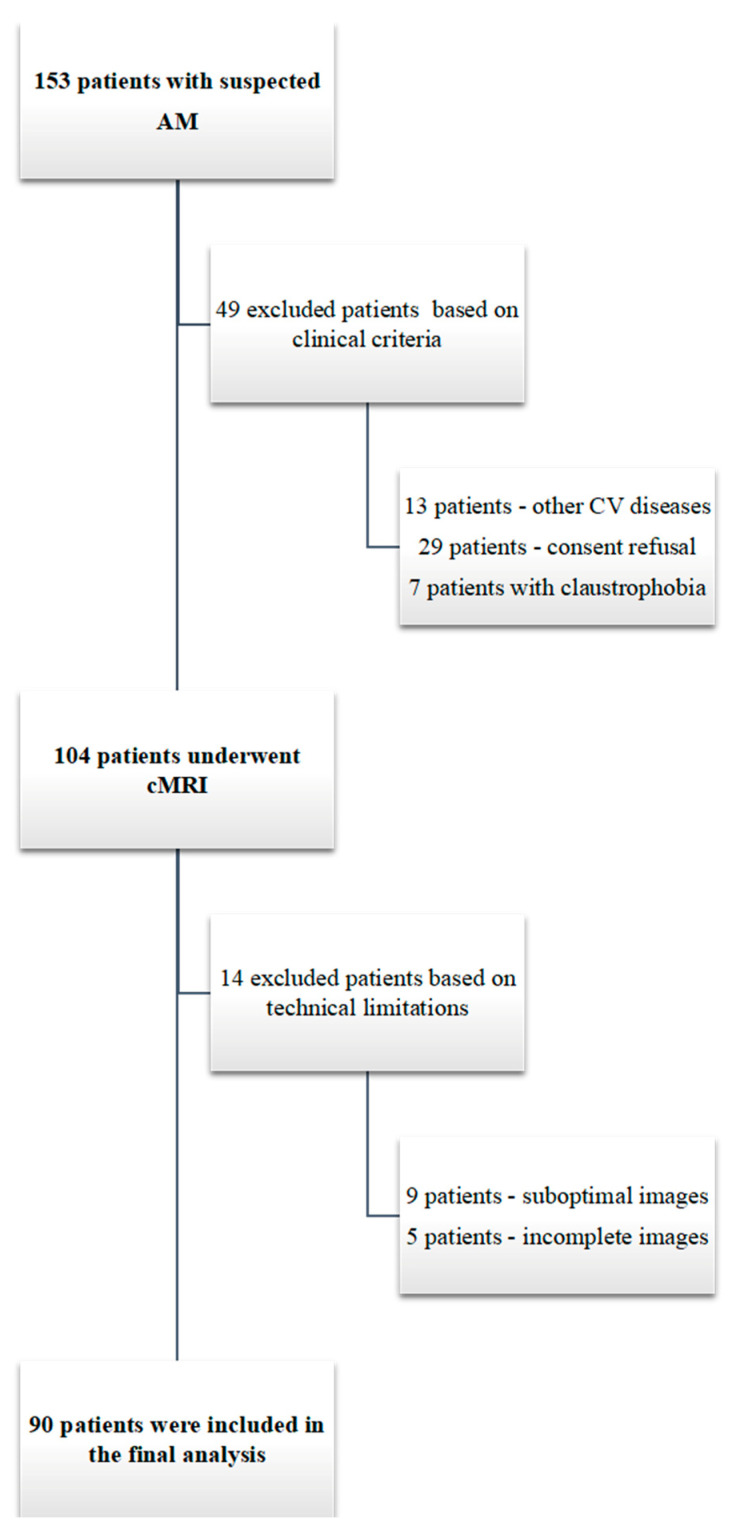
Flow diagram of patient selection in the study.

**Figure 3 diagnostics-14-01426-f003:**
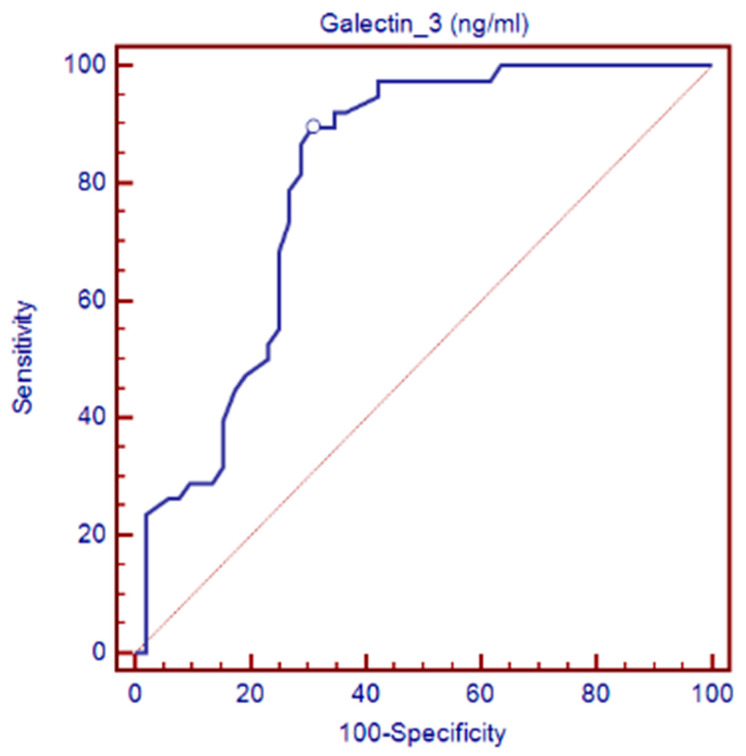
Characteristics of the ROC curve on the diagnostic capacity of galectin-3 (Gal-3) levels in the identification of myocardial fibrosis in patients with AM.

**Table 1 diagnostics-14-01426-t001:** Main characteristics of the patients included in the study.

General Data		
Age, years	35	
Male, %	76.6	
Female, %	23.3	
BMI, kg/m^2^	26.1	
HR, bpm	72	
SBP, mm Hg	125	
Symptoms	All patients (*n* = 90)	%
Chest pain	82	91.1
Dyspnea	54	60
Sweating	32	35.5
Syncope/lipothymia	8	8.8
Palpitations	23	25.5
Electrocardiogram (ECG)	All patients (*n* = 90)	%
ECG changes	83	92
ST elevation	23	25.5
Non-specific ST-T changes	67	74.4
Atrial fibrillation	6	6.6
LAB TEST	Average values	IQR
hs-troponin T	651 ng/L	123–9300 ng/L
hs-CRP	13.5 ng/mL	6.2–19.9 ng/mL
NT-proBNP	960 pg/mL	120–9893 pg/mL
Gal-3	13.5 ng/mL	7.5–17.9 ng/mL

BMI, body mass index; HR, heart rate; SBP, systolic blood pressure; hs-troponin T, high-sensibility troponin T; hs-CRP; high-sensibility C-reactive protein; NT-proBNP, N-terminal pro-B-type natriuretic peptide; Gal-3, galectin-3; and IQR, interquartile range.

**Table 2 diagnostics-14-01426-t002:** cMRI characteristics of study patients.

	Initial cMRI (*n* = 90)	cMRI at 6 Months (*n* = 90)	*p*-Value
Indexed EDV, average (SD), ml/m^2^	82.2 (19.2)	80.4 (17.1)	NS
Indexed ESV, average (SD), ml/m^2^	34.5 (17.4)	34.4 (13.4)	NS
LVEF, average (SD), %	58.8 (10.2)	60.1 (10.8)	NS
Segmentary kinetic abnormalities, *n* (%)	55 (61.1)	23 (25.5)	<0.001
LV mass, g/m^2^ (SD)	86 (19.2)	84 (17.3)	NS
T2 + edema, *n* (%)	84 (93.3)	34 (37.7)	<0.001
LGE +, *n* (%)	88 (97.7)	70 (77.7)	<0.05
-Myocardial localization—septum/lateral/anterior/inferior/ circumferential	35/12/26/11 /5	30/10/22/8/4	N/A
-Myocardial pattern— subepicardial/nodal/midmyocardial	52-12-24	47-08-19	N/A
LV-LGE, g	19.7 (10.5)	12.3 (6.7)	<0.05
LV-LGE/LV mass	26.1 (9.8)	20.7 (9.9)	<0.05
LGE without edema, *n* (%)	0	74 (82.2)	N/A
Edema without LGE, *n* (%)	8 (8.8)	0	N/A
T1 native mapping, ms	1123 ± 56	1037 ± 24	<0.01
T1 extended native mapping, *n* (%)	87 (96.6)	53 (58.8)	<0.05
Pericardial collection +, *n* (%)	43 (47.7)	23 (25.5)	<0.001

cMRI, cardiac magnetic resonance imaging; EDV, end-diastolic volume; ESV, end-systolic volume; LVEF, left ventricular ejection fraction; LV, left ventricle; LGE, late gadolinium enhancement; N/A, not applicable; NS, non-significance; and SD, standard deviation.

**Table 3 diagnostics-14-01426-t003:** Univariate and multivariate analyses by logarithmic regression.

	Unadjusted OR (95% CI)	*p*-Value	Adjusted OR (95% CI)	*p*-Value
Age, years	1.38 (1.15–1.66)	<0.01		
BMI, kg/m^2^	1.02 (0.93–1.11)	NS		
Chest pain	1.03 (0.99-1.81)	<0.01	1.00 (0.99–1.01)	NS
Dyspnea	1.04 (0.89–1.34)	<0.01	0.98 (0.89–1.18)	NS
Clinical heart failure	6.74 (2.72–17.27)	<0.001	3.44 (1.52–9.42)	<0.01
Troponin, ng/L	0.99 (0.99–1.01)	NS	1.23 (1.01–1.45)	0.05
NT-proBNP, pg/mL	1.42 (0.94–4.59)	0.001	1.01 (0.99–1.02)	NS
hs-CRP, ng/mL	5.79 (0.92–36.4)	<0.01	1.06 (0.94–1.14)	NS
Gal-3	2.67 (1.56–3.77)	<0.001	1.23 (1.09–2.11)	<0.01
LV kinetic abnormalities	0.96 (0.93–0.99)	NS		
ECG changes	0.99 (0.98–1.00)	<0.05	1.00 (0.99–1.01)	NS
Myocardial edema+	85.8 (6.03–318.74)	0.001	1.70 (1.14–209.3)	<0.001
LV-LGE, g	1.91 (1.25–2.97)	<0.001	1.27 (1.11–1.99)	<0.001
LV-LGE/LV mass	1.22 (1.18–1.40)	<0.01	1.00 (0.84–1.18)	NS
Medium-septal LGE pattern	1.19 (1.09–2.11)	<0.001	1.08 (0.92–1.56)	<0.01
Native T1 prolonged	1.10 (1.03–7.11)	<0.0001	0.97 (0.88–3.06)	<0.001

BMI, body mass index; NT-proBNP, N-terminal pro-B-type natriuretic peptide; hs-CRP, high-sensibility C-reactive protein; Gal-3, galectin-3; LV, left ventricle; LGE, late gadolinium enhancement; and NS, non-significance.

## Data Availability

The authors confirm that the data supporting the findings of this study are available within the article.

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
