# Peer review of "The Role of Magnetic Resonance Imaging in Risk Stratification of Patients with Acute Myocarditis"

_diagnostics, 2024, doi:10.3390/diagnostics14131426_

Round 1

Reviewer 1 Report

Comments and Suggestions for Authors

Popa A et al. have written an insightful paper on the role of cardiac magnetic resonance (CMR) in risk stratification for patients with acute myocarditis. The study involved 90 patients, all of whom underwent CMR within the first week after symptom onset and then again after six months.

The findings indicate that late gadolinium enhancement (LGE), myocardial edema, and prolonged native T1 are independent predictors of major adverse cardiac events (MACEs). Notably, the presence of LGE in the context of edema does not necessarily indicate established fibrosis, as it may resolve over time.

However, the study has limitations, including a small sample size and its retrospective design, which should be considered when interpreting the results.

Nevertheless, the findings are valuable and pave the way for future research.

Below are some points that should be considered to improve the article.

-        In the introductory section, I would suggest adding that CMR, through its ability to characterize myocardial tissue, enables precise diagnosis in patients who present clinically with acute coronary syndrome but have non-obstructed epicardial coronary arteries at the time of coronary angiography (citing PMID: 36889851, PMID: 37370978 and PMID: 37480903).

-        The authors should include chest pain and prodromal symptoms in Table 1.

-        In Table 3, both LGE (presumably included as a binary categorical variable) and LGE mass are included in the regression model. However, these two variables are collinear with each other, so it would be more appropriate to include only one of them. It is better to use LGE mass, as demonstrated in other studies (see PMID: 37480903).

-        DISCUSSION section: “At 6 months, ….. independent predictors of MACEs”. I suggest revising this section to simply summarize the key findings of the study as highlighted by the authors, which are further discussed later. This approach avoids repeating the list of differences observed between baseline and 6-month follow-up CMR.

-        DISCUSSION section: “The presence of LGE on cMRI appears to be associated with an improved prognosis”. The authors should provide further clarification on this statement, especially considering that their results clearly demonstrate how the presence of LGE is an independent predictor of MACE occurrence during the follow-up period.

Comments on the Quality of English Language

Minor editing of English language required

Author Response

Dear Doctor,

Thank you very much for revising the manuscript and for your detailed and useful comments. We have addressed each of them as follows.

Comments1: In the introductory section, I would suggest adding that CMR, through its ability to characterize myocardial tissue, enables precise diagnosis in patients who present clinically with acute coronary syndrome but have non-obstructed epicardial coronary arteries at the time of coronary angiography (citing PMID: 36889851, PMID: 37370978 and PMID: 37480903).

Response: We thank Reviewer for this comment and we agree. Accordingly, we have enhanced the text by citing the articles,  as required.

*CMRI, by characterizing myocardial tissue through LGE and abnormal T2 mapping, provides significant diagnostic and prognostic value for patients with acute coronary syndrome but non-obstructive coronary arteries during angiography (14-17).

Comments 2: The authors should include chest pain and prodromal symptoms in Table 1.

Response : Apologies for this oversight, we have updated Table 1 by adding  chest pain.

Comments 3: In Table 3, both LGE (presumably included as a binary categorical variable) and LGE mass are included in the regression model. However, these two variables are collinear with each other, so it would be more appropriate to include only one of them. It is better to use LGE mass, as demonstrated in other studies (see PMID: 37480903).

Response: Thank you for the suggestion, we have excluded the row with the LGE in Table 3.

Comments 4: DISCUSSION section: “At 6 months, ….. independent predictors of MACEs”. I suggest revising this section to simply summarize the key findings of the study as highlighted by the authors, which are further discussed later. This approach avoids repeating the list of differences observed between baseline and 6-month follow-up CMR.

Response: We thank the Reviewer for their comment. We have reformulated the text as follow:

At 6 months,  complete resolution (absence of edema and LGE) was achieved in a minority of the cases. Patients with LGE but without edema exhibited a more severe prognosis compared to those with persistent edema. Furthermore, patients showing increased extension of LGE on follow-up cardiac MRI had a worse prognosis than those with stable or reduced LGE. The mid-wall septal pattern of LGE and the presence of edema-free LGE were identified as independent predictors of major adverse cardiovascular events (MACEs).

Comments 5: DISCUSSION section: “The presence of LGE on cMRI appears to be associated with an improved prognosis”. The authors should provide further clarification on this statement, especially considering that their results clearly demonstrate how the presence of LGE is an independent predictor of MACE occurrence during the follow-up period.

Response: Apologies for this error, we have corrected the text, as follow:

The presence of LGE on cMRI appears to be associated with a worsened prognosis.

Reviewer 2 Report

Comments and Suggestions for Authors

Dear authors, 

I could not find any logical errors in the presentation and the approaches used. 

The paper is well written. The methodology is clear.

Patient selection and inclusion criteria were strictly followed. The analysis and results are explained clearly and thanks to the seriousness of the work they are absolutely reproducible. Your results underline the role of cMRI in the diagnosis and support the prognostic role of imaging. I may suggest you to propose a prognostic score for future investigations.

I would suggest just some minor corrections.

Figure 1 please make the description homogeneous between figures (a-e) ex. sequences and findings (arrow)

Table I

Please add in the patients result the unit of measure or percentage. It is not clear the way it is.

f.e. 

dispnea n,%

54 (60) it it’s the number 54 patient out of 60, you should add the percentage 

HR, average (SD) bpm 72(14,2)

 Table I describes the main characteristics of the patients included in your study.

You also add the number of cardiac CT and coronary angiography you performed in order to exclude coronary artery disease (exclusion criterion). I would suggest eliminating these two data form the table, that should summarize only your patient characteristics.

I would also suggest to divide the data (demographic-symptoms-la-ECG) otherwise it can be confusing.

Coronary artery disease exclusion: just please add a period in your patient selection result describing the examination you used (xxx coronary angiography, xxx cardiac CT).

Consider to add this citation

Liguori C, Tamburrini S, Ferrandino G, Leboffe S, Rosano N, Marano I. Role of CT and MRI in Cardiac Emergencies. Tomography. 2022 May 23;8(3):1386-1400. doi: 10.3390/tomography8030112. PMID: 35645398; PMCID: PMC9149871.

Author Response

Dear Doctor,

Thank you very much for revising the manuscript and for your detailed and useful comments. We have addressed each of them as follows.

Comments 1:  Figure 1 please make the description homogeneous between figures (a-e) ex. sequences and findings (arrow)

Response1:  Thank you for your indications, the pictures from Figure 1 were explain in full, as follow:

(a)Long axis 4 chambers -T2- weighted -STIR- hyper signal mid wall and subepicardial inferolateral regions of  LV= myocardial edema (green arrows). (b)Short axis-T2- weighted STIR-hyperintense subepicardial rim=myocardial hyperemia.(c) Short axis-Native T1 -mapping- prolonged T1 relaxation times in the inferolateral regions of the LV (1176 ± 40 msec)= interstitial fibrosis. (d) Long axis 4 chambers and (e) short axis -late gadolinium enhancement-patchy midwall and subepicardial LGE on inferolateral wall of the LV= permanent fibrosis (green arrows). (f) Sort axis-Native T2 mapping -prolonged T2 relaxation times in the inferolateral regions of the LV (60 ± 14 msec)=myocardial edema.

Comments 2: Table I. Please add in the patients result the unit of measure or percentage. It is not clear the way it is.f.e. dispnea n,% 54 (60) it it’s the number 54 patient out of 60, you should add the percentage HR, average (SD) bpm 72(14,2). I would also suggest to divide the data (demographic-symptoms-la-ECG) otherwise it can be confusing.

Response2:  We thank Reviewer for the suggestion, we have reformulate the table and we have add and split cells through which we have separated the data to be more clear.

Comments 3:  You also add the number of cardiac CT and coronary angiography you performed in order to exclude coronary artery disease (exclusion criterion). I would suggest eliminating these two data form the table, that should summarize only your patient characteristics.

Response3: Thank  you for this suggestion, we have eliminated the last two rows from Table 1.

Comments 4:  Coronary artery disease exclusion: just please add a period in your patient selection result describing the examination you used (xxx coronary angiography, xxx cardiac CT). 

Response4: Thank you for your indication. On page 3 we have add in the main text, from the total  number of patients how many were examined through angio-CT or coronary angiography- marked in red. As follow: Transthoracic echocardiography and cMRI were performed. Ischemic cardiovascular disease was excluded in all patients through either coronary angiography (72 patients) or coronary angioCT (18 patients).

Comments 5:  Consider to add this citation

Liguori C, Tamburrini S, Ferrandino G, Leboffe S, Rosano N, Marano I. Role of CT and MRI in Cardiac Emergencies. Tomography. 2022 May 23;8(3):1386-1400. doi: 10.3390/tomography8030112. PMID: 35645398; PMCID: PMC9149871.

Response5: We thank Reviewer for this suggestion. Accordingly, we have enhanced the text by citing the article on page 2 and 10.

Round 2

Reviewer 1 Report

Comments and Suggestions for Authors

Thank you to the authors for the revisions made, which I believe have enhanced the quality of the final manuscript. I have no further comments.